# Blood pressure control in patients aged above and below 75 years

**Hae Hyuk Jung**[ID]*

Department of Medicine, Kangwon National University Hospital, Kangwon National University School of Medicine, Chuncheon, South Korea

* haehyuk@kangwon.ac.kr

**Data Availability Statement:** Data cannot be shared publicly because of the sensitive nature of the data. Data are available from the National Health Information Database Institutional Data Access / Ethics Committee (contact via http://nhiss.

## Abstract

It remains unclear what the blood pressure target is and at which point in life it is appropriate for antihypertensive treatment. This study aimed to determine age-specific systolic blood pressure (SBP) targets. In a nationwide cohort of 296,470 hypertensive patients aged ≥75 years and a representative cohort of 259,028 hypertensives aged 45–74 years, multivariable-adjusted incidence rates of cardio-kidney composite events, overt dementia, and all-cause deaths were estimated across yearly-averaged on-treatment SBP levels according to age and the presence of 4 additional risk factors (diabetes, dyslipidemia, albuminuria, and smoking). For cardio-kidney events, on-treatment SBP showed positive curvilinear associations with higher risks at ≥135 mm Hg in most while an attenuated association in age ≥85 years. For overt dementia, SBP showed flat or slightly inverted associations in elderly while a small positive association in age 45–64 years. For all-cause mortality, SBP showed J-shaped associations having right-shifting tendency with age. For risk categories with ≥2, 1, and no additional risk factors, the respective mortality rate differences between SBP 145–154 mm Hg and 125–134 mm Hg were 4.6 (95% confidence interval [CI], 2.0 to 7.3), 1.2 (95% CI, -0.3 to 2.8), and 0.1 (95% CI, -1.4 to 1.8) per 1000 person-years in age ≥75 years and 2.9 (95% CI, 1.7 to 4.3), 0.7 (95% CI, 0.1 to 1.4), and 0.9 (95% CI, 0.2 to 1.6) per 1000 person-years in age 45–74 years. In conclusion, the BP target can be relaxed in very old patients and in elderly patients with few risk factors. However, strict BP control may be needed in patients with multiple risk factors even in those with advanced age.

## Introduction

Blood pressure (BP) increases with age, and the prevalence of hypertension is increasingly higher in older people. Further, other risk factors such as diabetes and hyperlipidemia are more prevalent in older people. Hypertension is associated with higher rates of cardiovascular disease, chronic kidney disease, and premature death, and BP-lowering treatment in patients with hypertension can improve outcomes. However, whether the benefit of BP reduction is comparable between older and younger patients is debated, and the appropriate BP targets for treatment among elderly patients with various comorbidities remain unclear.

nhis.or.kr) for researchers who meet the criteria for access to confidential data.

**Funding:** The author received no specific funding for this work.

**Competing interests:** The authors have declared that no competing interests exist.

A 2017 Cochrane review of randomized trials for adults aged 18 to 59 years with a baseline BP >140/90 mm Hg reported that antihypertensive treatment versus placebo or no treatment reduced cardiovascular mortality and morbidity but did not reduce all-cause mortality [1]. By comparison, in a 2019 Cochrane review of trials for adults 60 years or older with a baseline BP >140/90 mm Hg, active treatment versus placebo or no treatment reduced all-cause mortality among 60- to 79-year old people but did not among people 80 years or older [2]. These suggest that the benefit of antihypertensive treatment can vary depending on patient's age. However, the results derived from trials in highly selected patients are not readily generalizable to real-world practice. In contrast to the reviews of clinical trials, cohort or prospective observational studies showed positive continuous associations between BP levels and outcome risks across a wide range of age [3,4]. Although the participants could be representative of real-world populations, the findings might not reflect treatment effects because most of the observational studies did not account for treatment status. A few cohort studies separating on-treatment BP from untreated one showed J-shaped associations across on-treatment BP levels for outcome risks while showing linear associations across untreated levels [5,6]. It remains unclear what the BP target is and at which point in life it is appropriate for treatment to maximize its net benefit.

To determine age-specific BP targets for antihypertensive treatment, the incidences of cardio-kidney events, dementia, and deaths were estimated across on-treatment BP levels according to age in a nationwide cohort of patients 75 years or older and a representative cohort of patients aged 45 to 74 years in Korea.

## Methods

### Participants

This study was conducted from July to December 2021 in retrospective cohorts from the National Health Information Database. The database is managed by the National Health Insurance Service (NHIS) and covers data for the whole population of Korea [7]. The data used were de-identified prior to being provided. Access to the data was approved by the Institutional Review Board of Kangwon National University Hospital (IRB File No. KNHU-A-2021-04-017). The need for informed consent was waived because all data were fully anonymized before access.

This study enrolled all elderly participants of the nationwide health screening survey in 2009 or 2010 who were aged 75 years or older and collected their health screening records and NHIS reimbursement data from January 1, 2005 to December 31, 2019. From a total of 614,927 elderly participants, 17,749 who died before baseline (January 1, 2011), 25,214 with missing or outlier data, 31,838 who were diagnosed with major cancers, 4718 with an estimated glomerular filtration rate (eGFR) <15 ml/min/1.73 $m^2$ or end-stage kidney disease, and 41,238 who developed dementia before baseline were excluded. From the remaining 494,170 participants, 296,470 who had hypertension and on-treatment BP records at baseline were included in the final analysis (Fig 1A).

In addition, one-tenth of 45- to 74-year-old participants of the health screening survey were randomly selected to create a sample of the young-old and middle-aged populations. From the 976,985 participants, 3281 who died before baseline, 34,605 with missing or outlier data, 24,791 who were diagnosed with major cancers, 11,339 with an eGFR <15 ml/min/1.73 $m^2$ or end-stage kidney disease, and 5532 who developed dementia before baseline were excluded. From the remaining 897,437 participants, 259,028 who had hypertension and on-treatment BP records at baseline were included in the final analysis (Fig 1B).

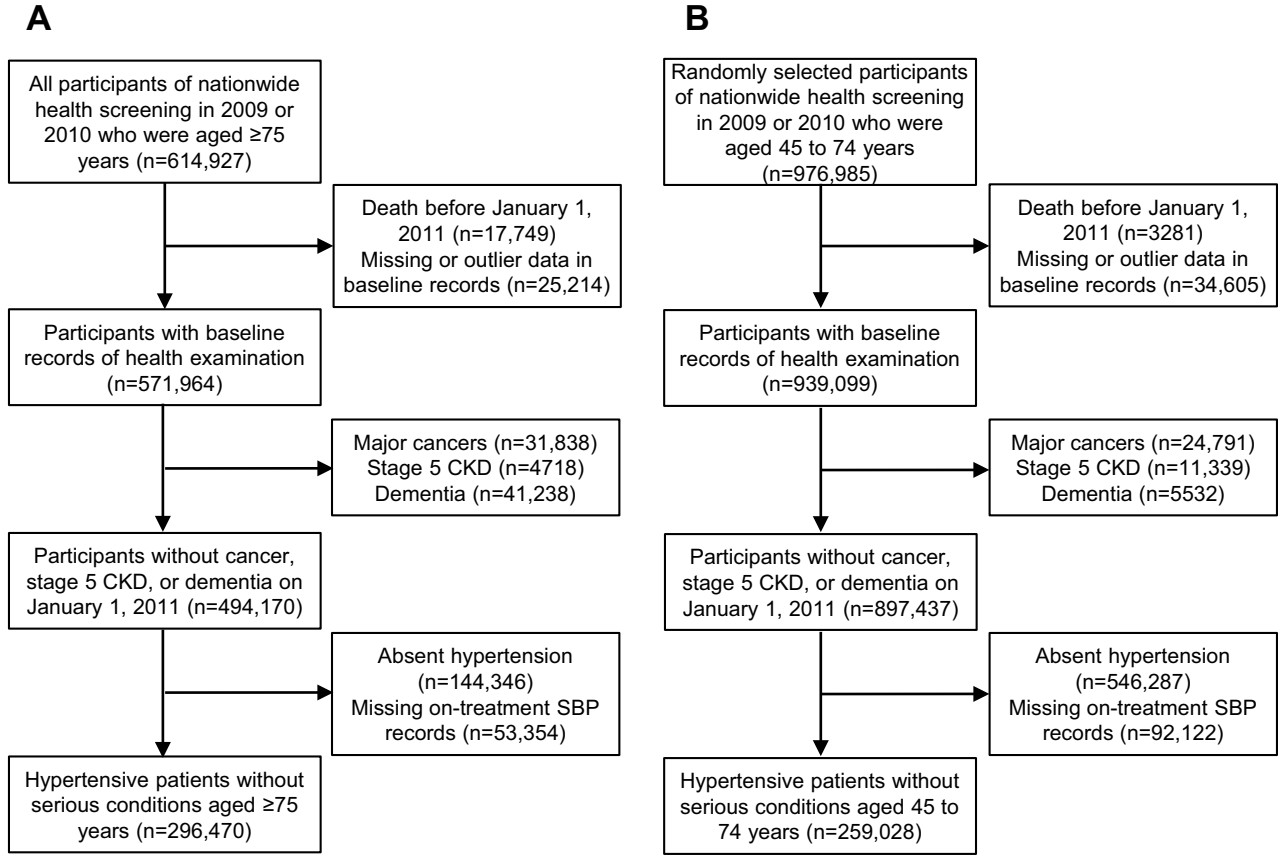

**Fig 1.** Flow chart of participants selection for hypertensive patients 75 years or older (A) and those aged 45 to 74 years (B).

## Fixed and time-varying covariates

Baseline covariates were determined using health screening and NHIS reimbursement data (S1 Table) between 2005 and 2010. Creatinine-based eGFR, dipstick albuminuria, total cholesterol, HDL cholesterol, fasting blood glucose, body mass index, waist circumference, income level, smoking status, alcohol consumption, frequency of physical exercise, medical history of cardiovascular disease, hypoglycemic treatment status, statin use status, and onset year of hypertension were determined and categorized referring to the publications (S1 Appendix).

Using baseline data, four additional risk factors (diabetes, dyslipidemia, albuminuria, and active smoking) were identified: diabetes as fasting glucose ≥126 mg/dL or prescription of hypoglycemic agents for ≥90 days per year; dyslipidemia as total cholesterol ≥240 mg/dL, HDL cholesterol <40 mg/dL, or prescription of statins for ≥90 days per year; albuminuria as urine dipstick albumin ≥1+ at least once or ≥trace twice; and active smoking as current smoking. Participants were classified into three risk categories by the number of additional risk factors (≥2, 1, or 0) present at baseline.

In each year of follow-up, antihypertensive treatment status was assessed and categorized into three groups (occasional, irregular, or regular medication): regular if the prescription was for >2/3 of the period from the initiation of medication to each year, irregular if the prescription was for 1/3 to 2/3 of each period, and occasional if the prescription was for <1/3 of each period.

During biennial health screenings, BP was measured using sphygmomanometers or oscillometric devices after at least 5-min rest. On-treatment and untreated BP values were separated as the risk threshold was markedly different depending on treatment status [5,6]. On-treatment BP was considered in the case that received antihypertensive agents for ≥90 days in the year of measurement. In each year, the separated systolic BP (SBP) values were averaged from 2005 and classified into eight categories (90–104, 105–114, 115–124, 125–134, 135–144, 145–154, 155–164, or 165–200 mm Hg).

## Outcome

The outcomes of interest were cardio-kidney composite outcome, overt dementia, and all-cause mortality. Cardio-kidney composite events consisted of hospitalized heart failure, critical atherscloerotic vascular events, cardiovascular death, and kidney failure. Hospitalized heart failure was identified as hospitalization with the primary diagnosis of heart failure (S1 Table). Critical atherscloerotic vascular events were identified as critical care unit admission or revascularization for acute coronary syndrome or acute ischemic stroke. Kidney failure was defined as end-stage kidney disease or death from chronic kidney disease. End-stage kidney disease was identified as dialysis for ≥90 days per year or kidney transplantation. Overt dementia consisted of moderate to severe dementia and death from dementia. Moderate to severe dementia was identified as the case that received donepezil, rivastigmine, galantamine, or memantine with the diagnosis of dementia. In Korea, cholinesterase inhibitors or memantine are reimbursed in the case with the MMSE score ≤26, the Clinical Dementia Rating ≥1, or the Global Deterioration Scale ≥3. All-cause deaths were confirmed by death certificates from Statistics Korea. Causes of death were identified by the primary cause of death on certificates. The first events of the outcomes were determined from January 1, 2011 through December 31, 2019, and censoring occurred with death or end of study.

## Statistical analysis

In time-dependent Cox models, multivariable-adjusted hazard ratios were estimated across on-treatment SBP levels. The models included yearly-averaged on-treatment SBP levels along with untreated levels and yearly-updated antihypertensive treatment status as time-varying covariates (S2 Table) and a continuous variable of baseline age and categorized variables of baseline eGFR, albuminuria, total cholesterol, HDL cholesterol, fasting blood glucose, body mass index, waist circumference, income level, smoking status, drinking amount, exercise frequency, history of cardiovascular disease, hypoglycemic treatment status, statin use status, onset year of hypertension, and sex as fixed covariates. Absolute risk was estimated as an incidence rate per person-years. The adjusted incidence rates and 95% CIs were calculated by multiplying the adjusted hazard ratios and their 95% CIs by a constant to make the sum of the products of the adjusted incidence rates and person-years in each SBP category equal the total number of observed events. In addition, spline regression was conducted with adjustment for the same covariates, to display associations of continuous SBP values and hazard ratios. The yearly-averaged on-treatment SBP was modelled using restricted cubic splines with 5 knots at 5, 27.5, 50, 72.5, and 95 percentiles.

The analyses were stratified by baseline age (oldest-old, ≥85 years; middle-old, 75–84 years; young-old, 65–74 years; and middle-age, 45–64 years) and further stratified by sex or the presence of risk factors. The interactions of SBP levels and age, sex, or risk cateogry were also analyzed. To test the proportional hazard assumption in Cox models, analyses were performed separately for the periods of 2011–2013, 2014–2016, and 2017–2019. To explore the influence of the inclusion or exclusion of patients with pre-existing heart disease on the results (e.g.,

reverse causality between BP and mortality), analyses were repeated after exclusion of participants with prior cardiovascular events. The analyses including baseline SBP rather than yearly-averaged levels were also performed to compare the risks of baseline levels with those of timely-updated levels. Statistical analyses were conducted using SAS software version 9.4 (SAS Institute). Data are presented as numbers and percentages, means and SDs, hazard ratios and 95% CIs, or incidence rates and 95% CIs. P values and 95% CIs were not adjusted for multiple testing.

## Results

### Baseline characteristics

The characteristics of 24,527 oldest-old (mean age, 87.2 years; 40.2% men), 271,943 middle-old (mean age, 78.5 years; 39.3% men), 94,918 young-old (mean age, 69.3 years; 44.4% men), and 164,110 middle-aged (mean age, 56.0 years; 51.4% men) participants are presented in Table 1. The older participants had lower eGFR levels, lower proportions of active smokers and heavy drinkers, and a higher proportion of sedentary persons.

### Age-stratified analyses

Over 9-year follow-up, in the oldest-old, middle-old, young-old, and middle-aged groups, cardio-kidney composite events were respectively noted in 23.8%, 14.5%, 6.7%, and 2.7% of participants, overt dementia respectively in 34.6%, 28.9%, 11.3%, and 1.4% of participants, and all-cause deaths respectively in 71.2%, 37.9%, 14.2%, and 3.8% of participants (S3–S8 Tables).

For cardio-kidney composite outcome, there were curvilinear associations with on-treatment SBP levels (Figs 2A and S1). Compared with a SBP of 125–134 mm Hg, the levels above 135 mm Hg were associated with a higher risk (Table 2) while the positive association was attenuated in the oldest-old group (P<0.001 for interaction between SBP and age group). In analyses stratified by sex (S2A and S3 Figs), the association curve was left-shifted in men relative to that in women (P = 0.001 for interaction) among participants 75 years or older while the association was similar between men and women (P = 0.53 for interaction) among 45- to 74-year-old participants.

For all-cause mortality, J-shaped associations were observed across on-treatment SBP levels (Figs 2C and S6), and the J-curves were right-shifted in older groups relative to those in younger groups (P<0.001 for interaction). In the oldest-old, middle-old, young-old, and middle-aged groups, the mortality rate differences between SBP levels of 145–154 mm Hg and 125–134 mm Hg were respectively estimated as -0.9 (95% CI, -6.5 to 5.0), 1.6 (95% CI, 0.6 to 2.6), 1.6 (95% CI, 0.6 to 2.7), and 1.2 (95% CI, 0.7 to 1.7) per 1000 person-years, (Table 3). In sex-stratified analyses (S2C and S7 Figs), the J-curve was slightly left-shifted in men relative to that in women (P = 0.014 for interaction) among older participants while the association did not differ between men and women (P = 0.50 for interaction) among younger participants.

### Risk-stratified analyses

In analyses stratified by the presence of risk factors (S9–S11 Tables), the association curve for SBP and cardio-kidney risk was left-shifted in risk category with ≥2 additional risk factors relative to that in categories with 1 or no additional risk factor (Figs 3A and S8) among both older (P = 0.001 for interaction) and younger (P = 0.010 for interaction) participants. For dementia risk, there was no consistent trend in associations with SBP across the risk categories (Figs 3B and S9).

**Table 1. Baseline characteristics of the study participants.**

| Characteristic | Age ≥85 Years | Age 75–84 Years | Age 65–74 Years | Age 45–64 Years |
|---|---|---|---|---|
| No. of participants | 24,527 | 271,943 | 94,918 | 164,110 |
| Age, mean (SD), yr | 87.2 (2.4) | 78.5 (2.4) | 69.3 (2.7) | 56.0 (5.3) |
| Men, no. (%) | 9,855 (40.2%) | 106,896 (39.3%) | 42,141 (44.4%) | 84,280 (51.4%) |
| Risk categories,* no. (%) | | | | |
| ≥2 additional risk factors | 4,002 (16.3%) | 56,330 (20.7%) | 22,359 (23.6%) | 38,532 (23.5%) |
| 1 additional risk factor | 9,923 (40.5%) | 116,114 (42.7%) | 40,862 (43.0%) | 67,748 (41.3%) |
| No additional risk factor | 10,602 (43.2%) | 99,499 (36.6%) | 31,697 (33.4%) | 57,830 (35.2%) |
| Onset year of hypertension, no. (%) | | | | |
| 2006 or less | 19,781 (80.6%) | 221,520 (81.5%) | 74,578 (78.6%) | 114,257 (69.6%) |
| 2007 or 2008 | 3,124 (12.7%) | 34,134 (12.6%) | 13,707 (14.4%) | 31,401 (19.1%) |
| 2009 or 2010 | 1,622 (6.6%) | 16,289 (6.0%) | 6,633 (7.0%) | 18,452 (11.2%) |
| Prior cardiovascular disease, no. (%) | 3,689 (15.0%) | 48,514 (17.8%) | 14,569 (15.3%) | 14,464 (8.8%) |
| Antihypertensive treatment, no. (%) | | | | |
| Occasional medication | 1,601 (6.5%) | 16,657 (6.1%) | 6,356 (6.7%) | 18,019 (11.0%) |
| Irregular medication | 2,937 (12.0%) | 24,183 (8.9%) | 7,598 (8.0%) | 15,731 (9.6%) |
| Regular medication | 19,989 (81.5%) | 231,103 (85.0%) | 80,964 (85.3%) | 130,360 (79.4%) |
| Hypoglycemic treatment, no. (%) | | | | |
| No or occasional medication | 20,879 (85.1%) | 218,068 (80.2%) | 74,412 (78.4%) | 137,988 (84.1%) |
| Irregular medication | 332 (1.4%) | 3,639 (1.3%) | 1,251 (1.3%) | 2,340 (1.4%) |
| Regular medication | 3,316 (13.5%) | 50,236 (18.5%) | 19,255 (20.3%) | 23,782 (14.5%) |
| Statin therapy, no. (%) | | | | |
| No or occasional medication | 21,270 (86.7%) | 215,325 (79.2%) | 70,772 (74.6%) | 127,129 (77.5%) |
| Irregular medication | 735 (3.0%) | 13,507 (5.0%) | 6,175 (6.5%) | 10,379 (6.3%) |
| Regular medication | 2,522 (10.3%) | 43,111 (15.9%) | 17,971 (18.9%) | 26,602 (16.2%) |
| SBP, mean (SD), mm Hg | 135.2 (14.8) | 135.0 (13.5) | 134.2 (12.8) | 132.3 (12.5) |
| FBG, mean (SD), mg/dl | 104.8 (25.7) | 104.9 (25.1) | 105.9 (25.8) | 105.7 (27.8) |
| Total cholesterol, mean (SD), mg/dl | 196.2 (32.9) | 198.2 (31.7) | 198.5 (30.8) | 199.8 (30.2) |
| HDL cholesterol, mean (SD), mg/dl | 51.4 (12.9) | 51.7 (12.7) | 52.1 (12.5) | 52.8 (12.4) |
| Body mass index, mean (SD), kg/m$^2$ | 23.0 (3.2) | 24.1 (3.1) | 24.9 (2.9) | 25.3 (3.0) |
| Waist circumference, mean (SD), cm | 82.8 (9.0) | 84.3 (8.3) | 85.2 (7.8) | 84.6 (8.1) |
| eGFR, mean (SD), ml/min/1.73 m$^2$ | 59.2 (15.7) | 66.4 (15.7) | 74.2 (15.3) | 83.5 (15.7) |
| Albuminuria, no. (%) | | | | |
| No albuminuria | 22,655 (92.4%) | 251,996 (92.7%) | 88,236 (93.0%) | 152,087 (92.7%) |
| Dipstick albumin 1+ or trace[†] | 1,227 (5.0%) | 12,493 (4.6%) | 4,118 (4.3%) | 7,389 (4.5%) |
| Dipstick albumin ≥2+[†] | 645 (2.6%) | 7,454 (2.7%) | 2,564 (2.7%) | 4,634 (2.8%) |
| Income level, no. (%) | | | | |
| Highest | 7,783 (31.7%) | 92,535 (34.0%) | 21,561 (22.7%) | 34,069 (20.8%) |
| High | 5,707 (23.3%) | 66,539 (24.5%) | 26,154 (27.6%) | 38,795 (23.6%) |
| Middle | 4,434 (18.1%) | 48,961 (18.0%) | 21,099 (22.2%) | 40,081 (24.4%) |
| Low | 3,348 (13.7%) | 36,080 (13.3%) | 15,226 (16.0%) | 33,084 (20.2%) |
| Lowest | 3,255 (13.3%) | 27,828 (10.2%) | 10,878 (11.5%) | 18,081 (11.0%) |
| Smoking, no. (%) | | | | |
| Never smoked | 19,652 (80.1%) | 211,343 (77.7%) | 69,353 (73.1%) | 103,069 (62.8%) |
| Former smoker | 3,391 (13.8%) | 39,136 (14.4%) | 15,210 (16.0%) | 31,868 (19.4%) |
| Current smoker | 1,484 (6.1%) | 21,464 (7.9%) | 10,355 (10.9%) | 29,173 (17.8%) |
| Physical exercise, no. (%) | | | | |
| <1 day/week | 11,603 (47.3%) | 99,357 (36.5%) | 26,191 (27.6%) | 35,993 (21.9%) |
| 1–2 days/week | 4,396 (17.9%) | 46,330 (17.0%) | 15,037 (15.8%) | 36,508 (22.2%) |

*(Continued)*

**Table 1.** (Continued)

| Characteristic | Age ≥85 Years | Age 75–84 Years | Age 65–74 Years | Age 45–64 Years |
|---|---|---|---|---|
| 3–4 days/week | 3,080 (12.6%) | 44,763 (16.5%) | 19,633 (20.7%) | 42,135 (25.7%) |
| ≥5 days/week | 5,448 (22.2%) | 81,493 (30.0%) | 34,057 (35.9%) | 49,474 (30.1%) |
| Drinking, no. (%) | | | | |
| Non drinking | 20,992 (85.6%) | 222,654 (81.9%) | 69,619 (73.3%) | 91,030 (55.5%) |
| 1–7 drinks/week | 2,078 (8.5%) | 26,240 (9.6%) | 11,715 (12.3%) | 28,725 (17.5%) |
| 8–28 drinks/week | 1,120 (4.6%) | 17,185 (6.3%) | 10,132 (10.7%) | 32,978 (20.1%) |
| ≥29 drinks/week | 337 (1.4%) | 5,864 (2.2%) | 3,452 (3.6%) | 11,377 (6.9%) |

* Participants were divided into 3 risk categories by the number of the risk factors present at baseline: i.e., ≥2, 1, or 0 of the 4 risk factors (diabetes, dyslipidemia, albuminuria, and smoking).

† The subgroups had dipstick albuminuria 1+ at least once or trace twice and dipstick albuminuria ≥2+ at least once during health screenings 2007 to 2010.

eGFR, estimated glomerular filtration rate; FBG, fasting blood glucose; HDL, high-density lipoprotein; SBP, systolic blood pressure.

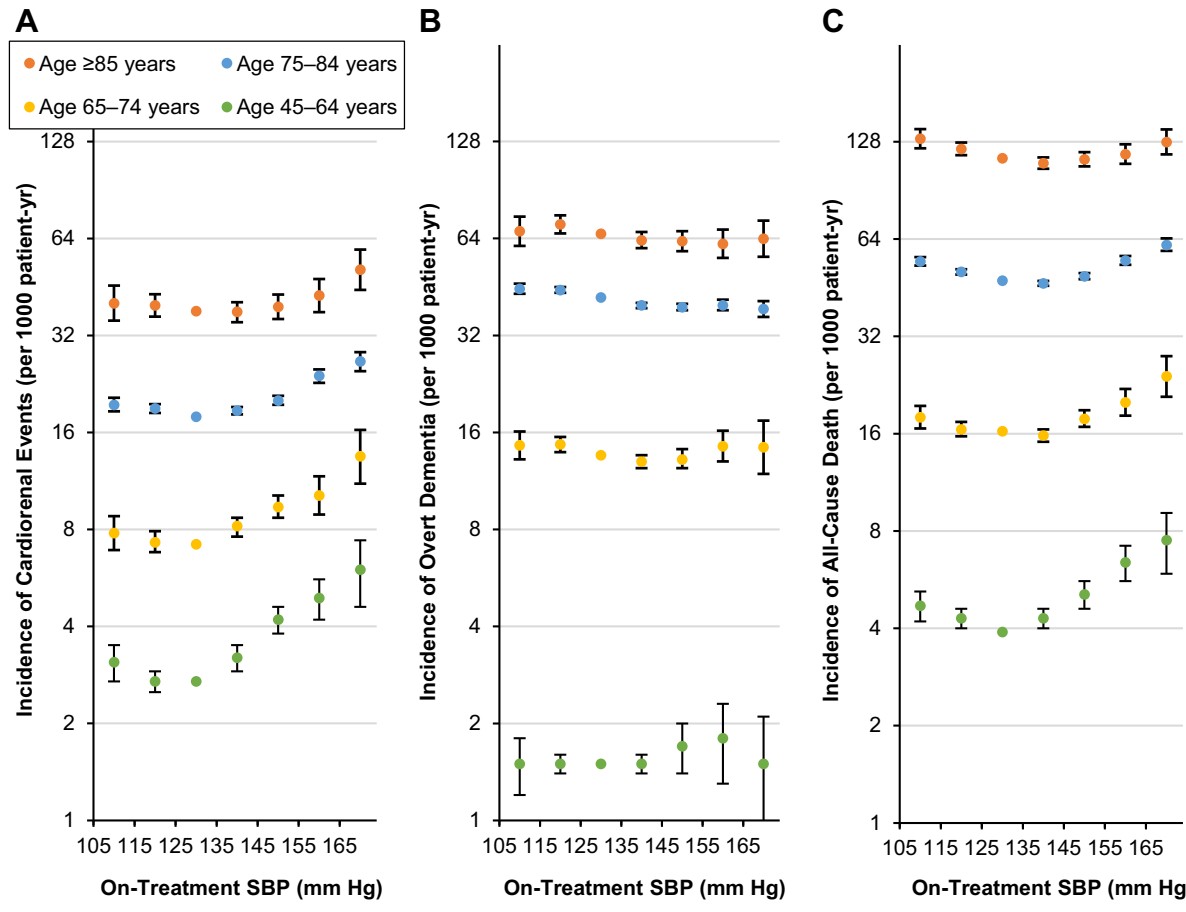

**Fig 2.** Adjusted incidence rates of cardio-kidney events (A), overt dementia (B), and all-cause death (C) according to age. The incidence rates and 95% CIs (error bars) in on-treatment SBP categories (90–104, 105–114, 115–124, 125–134, 135–144, 145–154, 155–164, or 165–200 mm Hg) were calculated by multiplying the adjusted hazard ratios and their 95% CIs by a constant to make the sum of the products of incidence rates and person-years in SBP categories equal the total number of observed events. The SBP category 90–104 mm Hg is excluded from the plots. The hazard ratios were adjusted for age, sex, onset year of hypertension, history of cardiovascular disease, antihypertensive, hypoglycemic, and statin treatment statuses, untreated SBP, fasting blood glucose, total and HDL cholesterols, estimated glomerular filtration rate, albuminuria, body mass index, waist circumference, income level, smoking status, exercise frequency, and drinking amount. The SBP 125–134 mm Hg was set as the reference. CI, confidence interval; SBP, systolic blood pressure.

**Table 2. Difference in cardio-kidney event rate compared with a SBP of 125–134 mm Hg according to age and the presence of risk factors\*.**

| Subgroup | SBP 115–124 mm Hg | SBP 135–144 mm Hg | SBP 145–154 mm Hg |
|---|---|---|---|
| | difference in incidence rate (95% CI), per 1000 person-years | | |
| Age category | | | |
| ≥85 years | 1.6 (-1.5 to 4.9) | -0.2 (-2.9 to 2.6) | 1.2 (-2.1 to 4.8) |
| 75–84 years | 1.1 (0.5 to 1.7) | 0.8 (0.3 to 1.3) | 2.2 (1.6 to 2.9) |
| 65–74 years | 0.1 (-0.4 to 0.7) | 0.9 (0.4 to 1.5) | 2.2 (1.4 to 3.0) |
| 45–64 years | 0.0 (-0.2 to 0.2) | 0.5 (0.2 to 0.7) | 1.5 (1.0 to 1.9) |
| Risk category in ≥75 years | | | |
| ≥2 additional risk factors | 1.4 (0.0 to 3.0) | 2.2 (0.9 to 3.6) | 5.2 (3.5 to 7.1) |
| 1 additional risk factor | 1.3 (0.5 to 2.2) | 0.2 (-0.5 to 1.0) | 1.6 (0.7 to 2.6) |
| No additional risk factor | 0.7 (-0.2 to 1.6) | 0.5 (-0.3 to 1.2) | 1.3 (0.3 to 2.3) |
| Risk category in 45–75 years | | | |
| ≥2 additional risk factors | -0.7 (-1.4 to 0.0) | 1.5 (0.8 to 2.3) | 3.6 (2.5 to 4.8) |
| 1 additional risk factor | -0.1 (-0.4 to 0.3) | 0.6 (0.2 to 0.9) | 1.4 (0.9 to 2.0) |
| No additional risk factor | 0.5 (0.1 to 0.8) | 0.2 (-0.1 to 0.5) | 0.8 (0.3 to 1.3) |

\* The incident rate difference was calculated from the incidence rates and 95% CIs which were calculated by multiplying the adjusted hazard ratios and their 95% CIs by a constant to make the sum of the products of incidence rates and person-years in SBP categories equal the total number of observed events. The SBP 125–134 mm Hg was set as the reference.

CI, confidence interval; SBP, systolic blood pressure.

**Table 3. Difference in all-cause death rate compared with a SBP of 125–134 mm Hg according to age and the presence of risk factors\*.**

| Subgroup | SBP 115–124 mm Hg | SBP 135–144 mm Hg | SBP 145–154 mm Hg |
|---|---|---|---|
| | difference in incidence rate (95% CI), per 1000 person-years | | |
| Age category | | | |
| ≥85 years | 7.7 (2.4 to 13.3) | -4.0 (-8.3 to 0.6) | -0.9 (-6.5 to 5.0) |
| 75–84 years | 3.1 (2.2 to 4.0) | -0.9 (-1.6 to -0.1) | 1.6 (0.6 to 2.6) |
| 65–74 years | 0.3 (-0.5 to 1.1) | -0.5 (-1.2 to 0.2) | 1.6 (0.6 to 2.7) |
| 45–64 years | 0.4 (0.1 to 0.7) | 0.4 (0.1 to 0.7) | 1.2 (0.7 to 1.7) |
| Risk category in ≥75 years | | | |
| ≥2 additional risk factors | 3.1 (0.8 to 5.5) | -1.0 (-2.9 to 1.0) | 4.6 (2.0 to 7.3) |
| 1 additional risk factor | 3.4 (2.0 to 4.8) | -1.2 (-2.3 to 0.0) | 1.2 (-0.3 to 2.8) |
| No additional risk factor | 3.4 (1.9 to 4.9) | -1.0 (-2.2 to 0.2) | 0.1 (-1.4 to 1.8) |
| Risk category in 45–75 years | | | |
| ≥2 additional risk factors | -0.1 (-0.9 to 0.8) | 0.4 (-0.4 to 1.3) | 2.9 (1.7 to 4.3) |
| 1 additional risk factor | 0.2 (-0.3 to 0.7) | -0.1 (-0.5 to 0.4) | 0.7 (0.1 to 1.4) |
| No additional risk factor | 0.9 (0.4 to 1.5) | 0.1 (-0.4 to 0.5) | 0.9 (0.2 to 1.6) |

\* The incident rate difference was calculated from the incidence rates and 95% CIs which were calculated by multiplying the adjusted hazard ratios and their 95% CIs by a constant to make the sum of the products of incidence rates and person-years in SBP categories equal the total number of observed events. The SBP 125–134 mm Hg was set as the reference.

CI, confidence interval; SBP, systolic blood pressure.

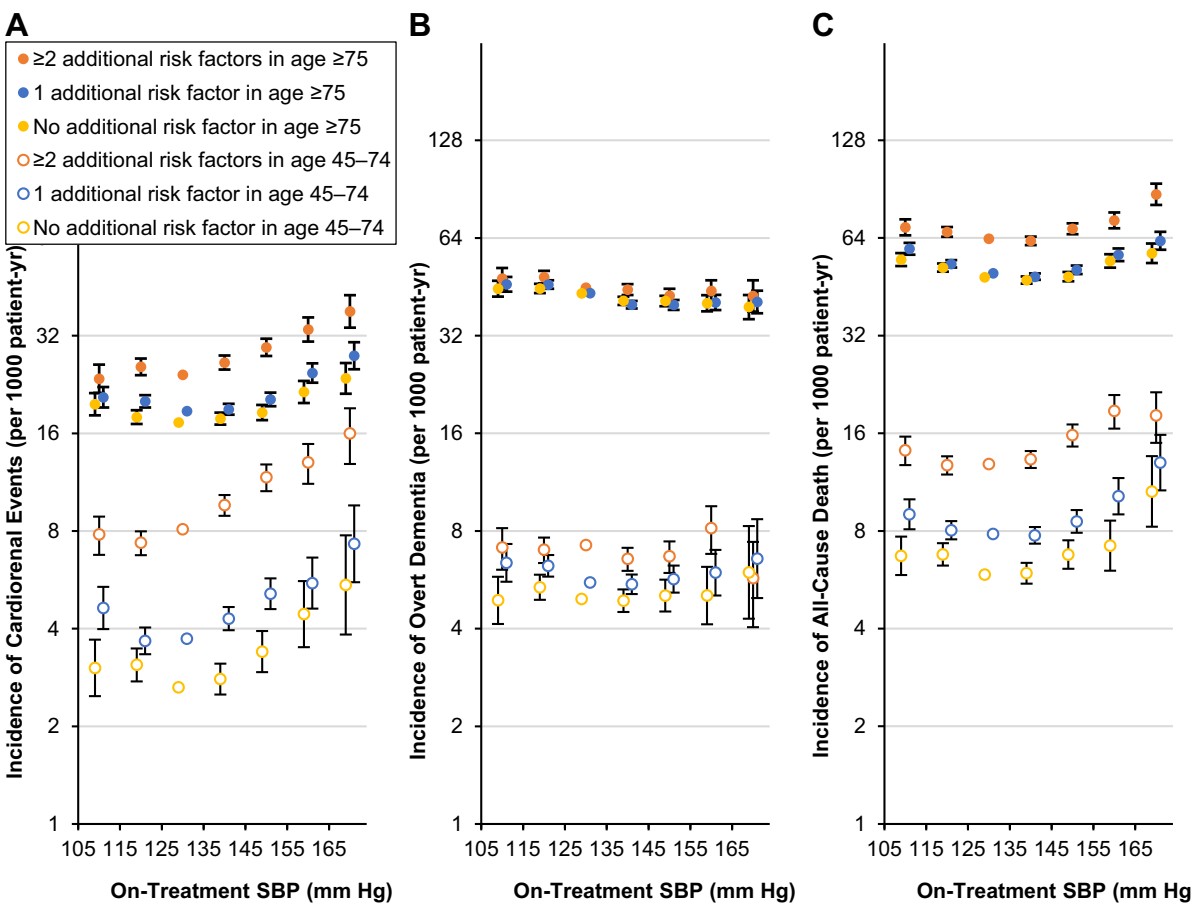

**Fig 3.** Adjusted incidence rates of cardio-kidney events (A), overt dementia (B), and all-cause death (C) according to the presence of risk factors. The incidence rates and 95% CIs (error bars) in on-treatment SBP categories (90–104, 105–114, 115–124, 125–134, 135–144, 145–154, 155–164, or 165–200 mm Hg) were calculated by multiplying the adjusted hazard ratios and their 95% CIs by a constant to make the sum of the products of incidence rates and person-years in SBP categories equal the total number of observed events. The SBP category 90–104 mm Hg is excluded from the plots. The hazard ratios were adjusted for age, sex, onset year of hypertension, history of cardiovascular disease, antihypertensive, hypoglycemic, and statin treatment statuses, untreated SBP, fasting blood glucose, total and HDL cholesterols, estimated glomerular filtration rate, albuminuria, body mass index, waist circumference, income level, smoking status, exercise frequency, and drinking amount. The SBP 125–134 mm Hg was set as the reference. CI, confidence interval; SBP, systolic blood pressure.

For all-cause mortality, the J-curve was left-shifted in risk categories with more risk factors (Figs 3C and S10) among both older (P = 0.010 for interaction) and younger (P = 0.024 for interaction) participants. In categories of ≥2, 1, and no additional risk factors, the respective mortality rate differences between SBP levels 145–154 mm Hg and 125–134 mm Hg were 4.6 (95% CI, 2.0 to 7.3), 1.2 (95% CI, -0.3 to 2.8), and 0.1 (95% CI, -1.4 to 1.8) per 1000 person-years among older and 2.9 (95% CI, 1.7 to 4.3), 0.7 (95% CI, 0.1 to 1.4), and 0.9 (95% CI, 0.2 to 1.6) per 1000 person-years among younger participants (Table 3).

## Additional analyses

When analyses were performed separately for the periods of 2011–2013, 2014–2016, and 2017–2019, the association curves for SBP and outcome risks were not apparently different across the time periods, indicating that there were no severe violations in the proportional hazard assumption (S11–S13 Figs). In analyses performed after exclusion of participants with prior cardiovascular events, the associations of on-treatment SBP and risks of dementia and

mortality were very similar to those in the primary analysis (S12–S14 Tables). When baseline SBP levels rather than yearly-averaged levels were introduced in Cox models, the associations between SBP and outcome risks were attenuated although the overall trends of right-shifted curves in older groups persisted (S14 Fig).

## Discussion

This cohort study in Korea for patients with hypertension who were aged above and below 75 years assessed outcome risks across on-treatment SBP levels according to age and risk status. In age-stratified analyses, the on-treatment SBP levels with the lowest risks for cardio-kidney events and all-cause mortality were higher in older patients. However, the absolute risk difference between inadequately lowered and optimally controlled BP levels was similar or even greater in older patients except for oldest-old patients. In risk-stratified analyses, the on-treatment SBP levels with the lowest risk were lower in patients with multiple risk factors than in those with no or few risk factors. Furthermore, the absolute risk difference between inadequately and optimally lowered BP was greater in patients with more risk factors. These findings would provide relevant information in the management of high BP for the elderly and middle-aged patients with various comorbidities.

In this study, cardio-kidney composite risk increased as the on-treatment SBP increased above 135 mm Hg in most age groups while the threshold was substantially right-shifted in the oldest-old group (Table 2). All-cause mortality increased as the SBP increased above 145 mm Hg in the middle-old and young-old groups while the threshold shifted to the right in the oldest-old group and to 135 mm Hg in the middle-aged group (Table 3). In a meta-analysis of trials for adults 60 years or older with a baseline SBP ≥160 mm Hg, treatment to a SBP target <150 mm Hg reduced all-cause mortality (absolute risk reduction [ARR] 0.58% per year for high-risk patients; risk ratio [RR] 0.90, 95% CI 0.83 to 0.98) [8]. In addition, a 2019 Cochrane review of randomized trials comparing active treatment with placebo or no treatment in adults 60 years or older with a baseline BP >140/90 mm Hg reported that antihypertensive treatment reduced all-cause mortality among 60- to 79-year-old patients (ARR 0.30% per year; RR 0.86, 95% CI 0.79 to 0.95) but did not among patients 80 years or older (ARR 0.19% per year; RR 0.97, 95% CI 0.87 to 1.10) while reducing cardiovascular event rates in both age groups [2]. The only randomized trial showing a reduction in all-cause mortality in patients 80 years or older used low doses of a thiazide and angiotensin-converting–enzyme inhibitor [9]. The findings of the present study are concordant with the results of clinical trials, suggesting that less aggressive treatment is a better approach in very old patients.

However, in contrast to the present findings, a 2017 Cochrane review of randomized trials for predominantly healthy adults aged 18 to 59 years with mild to moderate hypertension reported that antihypertensive treatment versus placebo or no treatment had little or no effect on all-cause mortality (ARR 0.03% per year; RR 0.94, 95% CI 0.77 to 1.13) while having a small absolute benefit to reduce cardiovascular mortality and morbidity (ARR 0.18% per year; RR 0.78, 95% CI 0.67 to 0.91) [1]. The Medical Research Council Trial of Mild Hypertension [10] contributed 84% of the total participants reviewed, with a mean age of 50 years, a mean baseline BP of 160/98 mmHg, and a mean follow-up duration of 5 years. In that trial, incomplete outcome data was at high risk of bias as participant participation was terminated in the event of non-fatal stroke or myocardial infarction, and the duration of follow-up was relatively short. In this age group, long-term trials would be needed to confirm BP-lowering effect on all-cause mortality.

Meanwhile, in a 2020 Cochrane review comparing lower BP targets (≤135/85 mm Hg) with standard targets (≤140/90 mm Hg), all-cause mortality did not differ between the lower and standard target groups (ARR 0.06% per year; RR 0.95, 95% CI 0.86 to 1.05) although cardiovascular

event rates were reduced in the intensive treatment group [11]. The only large-scale trial, the Systolic Blood Pressure Intervention Trial [12], showing a reduction in all-cause mortality in the lower versus standard target group reported substantially lower mortality rates among participants 75 years or older (1.8% per year vs. 2.6% per year; hazard ratio [HR] 0.67, 95% CI 0.49 to 0.91) [13] compared with the rates in the general US population (4.5% per year to 4.8% per year for the age 75 to 84 years in 2010s) [14] and in this nationwide Korean cohort for hypertension (4.7% per year to 6.1% per year for patients aged 75 to 84 years). The participants of the trial would not represent a general hypertensive population. BP-lowering to a target lower than standard target is expected to reduce cardio-kidney risk but not to reduce all-cause mortality in general populations with hypertension, as supported by this population-based cohort study.

In this study, the on-treatment SBP levels with the lowest risk for cardio-kidney and mortality outcomes were lower in patients with multiple versus few risk factors, and the absolute risk difference between inadequately and optimally lowered BP was greater in patients with more risk factors even among those with advanced age (Tables 2 and 3). Concordantly, in a randomized trial for Chinese patients aged 60 to 80 years comparing a SBP target 110–129 mm Hg with 130–149 mm Hg, intensive treatment reduced cardiovascular event rates among those with a 10-year Framingham risk ≥15% (ARR 0.51% per year; HR 0.66, 95% CI 0.50 to 0.86) but not among those with a 10-year risk <15% (ARR 0.03% per year; HR 0.96, 95% CI 0.62 to 1.49) [15]. Additionally, among two placebo-controlled trials of renin-angiotensin system inhibitor plus thiazide treatment, active treatment reduced all-cause mortality in a trial for patients (mean age 66 years, mean baseline BP 145/81 mm Hg) with diabetes and at least one additional risk factor (ARR 0.26% per year; HR 0.86, 95% CI 0.75 to 0.98) [16] but not in the trial for participants (mean age 66 years, mean baseline BP 138/82 mm Hg) at intermediate risk (ARR 0.02% per year; HR 0.98, 95% CI 0.84 to 1.14) [17]. The benefit of BP reduction appears to be greater in patients at higher-risk, and the present study provides additional evidence that the greater benefit in higher-risk patients persists even among those with advanced age.

This study has several limitations. First, BP was measured in hospitals or medical centers using sphygmomanometers or oscillometric devices. The office-based BP could be affected by multiple factors and could not exclude the possible existence of white-coat effect. Next, analyses did not include treatment-related adverse events or symptoms. Although all-cause mortality can be a surrogate for net effect, future studies on adverse treatment effects will be needed to assess net benefit including quality of life. Furthermore, the analysis did not address frailty due to lack of the data in the NHID, although the variable of physical exercise was included in the models. Appropriate BP targets can be different according to the status of frailty in elderly patients. Moreover, vascular dementia and Alzheimer's disease were not separated in outcome measures although dementia diagnoses were verified using reimbursement records on the prescription of cholinesterase inhibitors or memantine. There remains a possibility that BP-related dementia risk varies depending on dementia subtypes. Besides, as the study included residents in Korea, caution is required when applying the findings to populations with other nationalities or ethnicities. Finally, given the observational nature, there could be bias from residual confounding or reverse causation. Large-scale randomized trials including older and younger patients are required to determine BP targets for each age group more convincingly, and pooled analyses of individual patient data from clinical trials will be needed to determine those among patients with various comorbidities.

## Conclusion

The findings of this study support an individualized approach of less aggressive BP-lowering in very old patients or in elderly patients with few risk factors and strict BP control in patients

with multiple risk factors even in those with advanced age. Further studies investigating treatment-related adverse events or quality of life, long-term randomized trials including a more representative patient sample, and meta-analyses of individual patient data from clinical trials will be needed to identify subgroups of patients who are likely to benefit from a specific BP target or treatment.

## Supporting information

**S1 Appendix. Covariates.**
(DOCX)

**S1 Table. NHIS codes for patient management and diagnosis.**
(DOCX)

**S2 Table. Time-lagged covariates for subsequent years.**
(DOCX)

**S3 Table. Adjusted hazard ratios and incidence rates of cardio-kidney events across on-treatment systolic blood pressure levels according to age.**
(DOCX)

**S4 Table. Adjusted hazard ratios and incidence rates of cardio-kidney events across on-treatment systolic blood pressure levels according to age and sex.**
(DOCX)

**S5 Table. Adjusted hazard ratios and incidence rates of overt dementia across on-treatment systolic blood pressure levels according to age.**
(DOCX)

**S6 Table. Adjusted hazard ratios and incidence rates of overt dementia across on-treatment systolic blood pressure levels according to age and sex.**
(DOCX)

**S7 Table. Adjusted hazard ratios and incidence rates of all-cause death across on-treatment systolic blood pressure levels according to age.**
(DOCX)

**S8 Table. Adjusted hazard ratios and incidence rates of all-cause death across on-treatment systolic blood pressure levels according to age and sex.**
(DOCX)

**S9 Table. Adjusted hazard ratios and incidence rates of cardio-kidney events across on-treatment systolic blood pressure levels according to the presence of risk factors.**
(DOCX)

**S10 Table. Adjusted hazard ratios and incidence rates of overt dementia across on-treatment systolic blood pressure levels according to the presence of risk factors.**
(DOCX)

**S11 Table. Adjusted hazard ratios and incidence rates of all-cause death across on-treatment systolic blood pressure levels according to the presence of risk factors.**
(DOCX)

**S12 Table. Adjusted hazard ratios and incidence rates of adverse events according to age after exclusion of participants with prior cardiovascular events.**
(DOCX)

**S13 Table. Adjusted hazard ratios and incidence rates of overt dementia according to risk status after exclusion of participants with prior cardiovascular events.**
(DOCX)

**S14 Table. Adjusted hazard ratios and incidence rates of all-cause death according to risk status after exclusion of participants with prior cardiovascular events.**
(DOCX)

**S1 Fig. Association of on-treatment systolic blood pressure and cardio-kidney risk according to age.**
(PDF)

**S2 Fig. Adjusted incidence rates of cardio-kidney events, overt dementia, and all-cause death according to age and sex.**
(PDF)

**S3 Fig. Association of on-treatment systolic blood pressure and cardio-kidney risk according to age and sex.**
(PDF)

**S4 Fig. Association of on-treatment systolic blood pressure and dementia risk according to age.**
(PDF)

**S5 Fig. Association of on-treatment systolic blood pressure and dementia risk according to age and sex.**
(PDF)

**S6 Fig. Association of on-treatment systolic blood pressure and all-cause mortality according to age.**
(PDF)

**S7 Fig. Association of on-treatment systolic blood pressure and all-cause mortality according to age and sex.**
(PDF)

**S8 Fig. Association of on-treatment systolic blood pressure and cardio-kidney risk according to the presence of risk factors.**
(PDF)

**S9 Fig. Association of on-treatment systolic blood pressure and dementia risk according to the presence of risk factors.**
(PDF)

**S10 Fig. Association of on-treatment systolic blood pressure and all-cause mortality according to the presence of risk factors.**
(PDF)

**S11 Fig. Hazard ratios of cardio-kidney events for the periods of 2011–2013, 2014–2016, and 2017–2019.**
(PDF)

**S12 Fig. Hazard ratios of overt dementia for the periods of 2011–2013, 2014–2016, and 2017–2019.**
(PDF)

**S13 Fig. Hazard ratios of all-cause mortality for the periods of 2011–2013, 2014–2016, and 2017–2019.**
(PDF)

**S14 Fig. Adjusted incidence rates of cardio-kidney events, overt dementia, and all-cause death across baseline on-treatment systolic blood pressure levels.**
(PDF)

## Author Contributions

**Conceptualization:** Hae Hyuk Jung.

**Data curation:** Hae Hyuk Jung.

**Formal analysis:** Hae Hyuk Jung.

**Investigation:** Hae Hyuk Jung.

**Methodology:** Hae Hyuk Jung.

**Project administration:** Hae Hyuk Jung.

**Resources:** Hae Hyuk Jung.

**Validation:** Hae Hyuk Jung.

**Visualization:** Hae Hyuk Jung.

**Writing – original draft:** Hae Hyuk Jung.

**Writing – review & editing:** Hae Hyuk Jung.

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
