## [Decision Letter · Decision Letter 0]

2 Nov 2023

PONE-D-23-27216Blood Pressure Control in Patients Aged Above and Below 75 YearsPLOS ONE

Dear Dr. Jung,

Thank you for submitting your manuscript to PLOS ONE. After careful consideration, we feel that it has merit but does not fully meet PLOS ONE’s publication criteria as it currently stands. Therefore, we invite you to submit a revised version of the manuscript that addresses the points raised during the review process.

We look forward to receiving your revised manuscript.

Kind regards,

Masaki Mogi

Academic Editor

PLOS ONE

Journal Requirements:

**Additional Editor Comments:**

The manuscript is necessary to be minor-revisioned according to the Reviewer's comments.

Reviewers' comments:

Reviewer's Responses to Questions

**Comments to the Author**

1. Is the manuscript technically sound, and do the data support the conclusions?

Reviewer #1: Yes

2. Has the statistical analysis been performed appropriately and rigorously? 

Reviewer #1: Yes

3. Have the authors made all data underlying the findings in their manuscript fully available?

Reviewer #1: Yes

4. Is the manuscript presented in an intelligible fashion and written in standard English?

Reviewer #1: Yes

5. Review Comments to the Author

Reviewer #1: PONE-D-23-27216

This paper reported the association of BP control level with cardio-renal mortality and development of dementia in each stratified age. The author concluded that there was little benefit for cardiovascular protection by strict BP management in elderly hypertensive patients. This issue should be handled carefully, especially in clinical settings. The reviewer has several comments.

1. The author used a big data from national insurance data. The methods of BP measurement might be not same in each patient. The authors should mention this limitation for BP measurements and existence of white-coat effect.

2. In international guideline, the target BP level is decided by the status of frailty in each elderly patient. If elderly patient is not frail but keep good activity, strict BP management should be option of hypertension treatment. If the author has the data for frailty, this variable should be included in analysis.

3. The reviewer recommends the stratified analysis for CKD, diabetes, and pre-history of cardiovascular disease. In the CKD patient, target range of BP is different.

4. The association between BP and development of heart failure is important. The author should add the analysis for each outcome of cardiovascular disease, i.e., stroke, coronary artery disease, and heart failure.

5. How about diastolic BP level and outcomes in age-stratified analysis?

6. PLOS authors have the option to publish the peer review history of their article (what does this mean?). If published, this will include your full peer review and any attached files.

Reviewer #1: No

---

## [Author Response · Author response to Decision Letter 0]

10 Dec 2023

Ref.: Ms. No. PONE-D-23-27216

Blood Pressure Control in Patients Aged Above and Below 75 Year

I thank the reviewer for his helpful comments. The manuscript has benefited from the comments. The manuscript has been rechecked and the necessary changes have been made in accordance with the reviewer’s suggestions. The responses to all comments have been prepared and are itemized below, in addition to having been implemented in the manuscript, as indicated. 

Review comments:

Reviewer #1: 

This paper reported the association of BP control level with cardio-renal mortality and development of dementia in each stratified age. The author concluded that there was little benefit for cardiovascular protection by strict BP management in elderly hypertensive patients. This issue should be handled carefully, especially in clinical settings. The reviewer has several comments.

1. The author used a big data from national insurance data. The methods of BP measurement might be not same in each patient. The authors should mention this limitation for BP measurements and existence of white-coat effect.

RESPONSE:, In this study, BP was measured during biennial health screenings using sphygmomanometers or oscillometric devices after at least 5-min rest. As you pointed out, BP obtained from routine office measurement is generally higher than that obtained via ambulatory or home measurement and cannot exclude white-coat effect. The Limitation section has been revised including this limitation. 

2. In international guideline, the target BP level is decided by the status of frailty in each elderly patient. If elderly patient is not frail but keep good activity, strict BP management should be option of hypertension treatment. If the author has the data for frailty, this variable should be included in analysis.

RESPONSE: This analysis could not address frailty due to lack of the data in the NHID although the variable of physical exercise was included in the models. The manuscript (Limitation section) has been revised including the limitation. 

3. The reviewer recommends the stratified analysis for CKD, diabetes, and pre-history of cardiovascular disease. In the CKD patient, target range of BP is different.

RESPONSE: At first, I regret that the additional analyses cannot be performed now because the date for accessing data has expired. However, even if possible, the stratified analyses in each age group would not provide reliable results as the number of participants in each group (e.g., CKD patients 85 years or older) was not sufficient. 

I have previously performed the analysis in a nationwide cohort including 359,492 CKD patients who had received antihypertensives regularly. In that analysis, the risk thresholds of on-treatment systolic BP, above which overall risk increased significantly, are around 130 mm Hg in CKD patients with proteinuria and 140 mm Hg in patients with no proteinuria (Nephrol Dial Transplant. 2022;37(6):1088). The results support that the systolic target for patients with CKD and proteinuria is ≤130 mm Hg, and the target for patients with no proteinuria is ≤140 mm Hg. 

In analyses performed after exclusion of participants with prior cardiovascular events, the associations of on-treatment SBP and risks of dementia and mortality were very similar to those in the primary analysis as shown in the Results (Additional analyses subsection) of the present manuscript (Tables S12-S14). I believe that the stratified analysis in patients with prior cardiovascular disease, even if possible, would not provide reliable results due to the insufficient number of participants.

4. The association between BP and development of heart failure is important. The author should add the analysis for each outcome of cardiovascular disease, i.e., stroke, coronary artery disease, and heart failure.

RESPONSE: As mentioned above, I regret that the analysis cannot be performed now because the expiration date for accessing data has passed. 

In a previous study conducted by me and colleagues (Hypertension. 2018;71(6):1047), among active users of antihypertensives, the risks for coronary, cerebrovascular, and renal events were lowest at systolic BPs of 125–<145 mm Hg, 115–<125 mm Hg, and <115 mm Hg, respectively, and at diastolic BPs of 75–<95 mm Hg, 65–<85 mm Hg, and <75 mm Hg, respectively. 

5. How about diastolic BP level and outcomes in age-stratified analysis?

RESPONSE: In the study mentioned above (Hypertension. 2018;71(6):1047), J-curves were also noted for diastolic BP and the J-curves for systolic and diastolic BP had similar shapes. In my opinion, as the wide pulse pressure is commonly found in elderly patients and is associated with poor outcomes, the diastolic BP levels with the lowest risks would be more right-shifted in older patients compared with the systolic levels. I regret that this paper could not show the results clearly.

---

## [Decision Letter · Decision Letter 1]

27 Dec 2023

Blood Pressure Control in Patients Aged Above and Below 75 Years

PONE-D-23-27216R1

Dear Dr. Jung,

We’re pleased to inform you that your manuscript has been judged scientifically suitable for publication and will be formally accepted for publication once it meets all outstanding technical requirements.

Kind regards,

Masaki Mogi

Academic Editor

PLOS ONE

Additional Editor Comments (optional):

Reviewers' comments:

Reviewer's Responses to Questions

**Comments to the Author**

1. If the authors have adequately addressed your comments raised in a previous round of review and you feel that this manuscript is now acceptable for publication, you may indicate that here to bypass the “Comments to the Author” section, enter your conflict of interest statement in the “Confidential to Editor” section, and submit your "Accept" recommendation.

Reviewer #1: All comments have been addressed

2. Is the manuscript technically sound, and do the data support the conclusions?

Reviewer #1: Yes

3. Has the statistical analysis been performed appropriately and rigorously? 

Reviewer #1: Yes

4. Have the authors made all data underlying the findings in their manuscript fully available?

Reviewer #1: Yes

5. Is the manuscript presented in an intelligible fashion and written in standard English?

Reviewer #1: Yes

6. Review Comments to the Author

Reviewer #1: This observational study may provide the pathological impact of BP in elderly patients. The authors responded the reviewer's comments carefully.

7. PLOS authors have the option to publish the peer review history of their article (what does this mean?). If published, this will include your full peer review and any attached files.

Reviewer #1: No

---

## [Editor Report · Acceptance letter]

24 Jan 2024

PONE-D-23-27216R1 

PLOS ONE

Dear Dr. Jung, 

I'm pleased to inform you that your manuscript has been deemed suitable for publication in PLOS ONE. Congratulations! Your manuscript is now being handed over to our production team.

Kind regards, 

on behalf of

Dr. Masaki Mogi 

Academic Editor

PLOS ONE